# Metaverse

**Stylianos Mystakidis** [1,2] 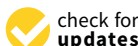

1  School of Natural Sciences, University of Patras, 26504 Patras, Greece; smyst@upatras.gr
2  School of Humanities, Hellenic Open University, 26335 Patras, Greece

**Definition:** The Metaverse is the post-reality universe, a perpetual and persistent multiuser environment merging physical reality with digital virtuality. It is based on the convergence of technologies that enable multisensory interactions with virtual environments, digital objects and people such as virtual reality (VR) and augmented reality (AR). Hence, the Metaverse is an interconnected web of social, networked immersive environments in persistent multiuser platforms. It enables seamless embodied user communication in real-time and dynamic interactions with digital artifacts. Its first iteration was a web of virtual worlds where avatars were able to teleport among them. The contemporary iteration of the Metaverse features social, immersive VR platforms compatible with massive multiplayer online video games, open game worlds and AR collaborative spaces.

**Keywords:** metaverse; mixed reality; virtual reality; augmented reality; extended reality; virtual worlds; multiuser virtual environments



## 1. Introduction

Computer Science innovations play a major role in everyday life as they change and enrich human interaction, communication and social transactions. From the standpoint of end users, three major technological innovation waves have been recorded centered around the introduction of personal computers, the Internet and mobile devices, respectively. Currently, the fourth wave of computing innovation is unfolding around spatial, immersive technologies such as Virtual Reality (VR) and Augmented Reality (AR) [1]. This wave is expected to form the next ubiquitous computing paradigm that has the potential to transform (online) education, business, remote work and entertainment. This new paradigm is the Metaverse. The word Metaverse is a closed compound word with two components: Meta (Greek prefix meaning post, after or beyond) and universe. In other words, the Metaverse is a post-reality universe, a perpetual and persistent multiuser environment merging physical reality with digital virtuality. Regarding online distance education, Metaverse has the potential to remedy the fundamental limitations of web-based 2D e-learning tools.

Education is one crucial field for society and economy where core implementation methods remain unchanged and orbiting around content transmission, classrooms and textbooks despite numerous technological innovations [2]. Currently, there is an intense race to construct the infrastructure, protocols and standards that will govern the Metaverse. Large corporations are striving to construct their closed, proprietary hardware and software ecosystems so as to attract users and become the de facto Metaverse destination. Different systemic approaches and diverging strategies collide around concepts such as openness and privacy. The outcome of this race will determine the level of users' privacy rights as well as whether the Metaverse will be inclusive to students and school pupils. Both issues have important implications for education as they will determine if the Metaverse can become mainstream in e-learning. The aim of this article is to raise awareness about the origin and the affordances of the Metaverse, so as to formulate a unified vision for meta-education, Metaverse-powered online distance education.

For this purpose, this article is structured as follows: Definitions of the key concepts are presented in Section 2. The limitations of two-dimensional learning environments are

summarized in Section 3. A brief historical account of virtual media and VR technology is illustrated in Section 4. Next, in Section 5, the significance of virtual worlds and VR for education is presented. Section 6 is dedicated to contemporary Metaverse development, Meta-education and innovative applications. The conclusions are in the final Section 7.

## 2. Extended, Virtual, Augmented and Mixed Reality

Extended Reality or Cross Reality (XR) is an umbrella term that includes a series of immersive technologies; electronic, digital environments where data are represented and projected. XR includes Virtual Reality (VR), Augmented Reality (AR) and Mixed Reality (MR) [3]. In all the above-mentioned XR facets, humans observe and interact in a fully or partially synthetic digital environment constructed by technology.

VR is an alternate, completely separate, digitally created, artificial environment. Users feel in VR that they are immersed, located in a different world and operate in similar ways just like in the physical surroundings [4]. With the help of specialized multisensory equipment such as immersion helmets, VR headsets and omnidirectional treadmills, this experience is amplified through the modalities of vision, sound, touch, movement and the natural interaction with virtual objects [5,6].

AR adopts a different approach towards physical spaces; it embeds digital inputs, virtual elements into the physical environment so as to enhance it [7]. It spatially merges the physical with the virtual world [8]. The end outcome is a spatially projected layer of digital artifacts mediated by devices, e.g., smart phones, tablets, glasses, contact lenses or other transparent surfaces [9]. Moreover, AR can also be implemented in VR headsets with pass-through mode capability by displaying input from integrated camera sensors.

MR is a more complex concept and its definition has fluctuated across time, reflecting the contemporary technological trends and dominant linguistic meanings and narratives [10]. MR is sometimes represented as an advanced AR iteration in the sense that the physical environment interacts in real time with the projected digital data [10]. For instance, a scripted non-player character in an MR game would recognize the physical surroundings and hide behind under a desk or behind a couch. Similar to VR, MR requires special glasses. However, for the purpose of this article, we accept the conception of MR as any combination of AR and VR as well as intermediate variations such as augmented virtuality [3]. The rationale behind this decision is the long-term technological evolution and maturation of AR to include interactive affordances. Therefore, AR and VR remain the two fundamental technologies and MR their combination.

To comprehend and visualize how these immersive technologies interact with the environment, we point to Milgram and Kishino's one-dimensional reality–virtuality continuum [3]. This continuum is illustrated as a straight line with two ends. On the left extremum of the line there is the natural, physical environment. The right end marks the fully artificial, virtual environment that the user experiences instead of the physical one. Hence, AR is near the left end of the spectrum while VR occupies the right extremum. MR is a superset of both.

*Multimodal Metaverse Interactions*

The Metaverse is based on technologies that enable multisensory interactions with virtual environments, digital objects and people. The representational fidelity of the XR system is enabled by stereoscopic displays that are able to convey the perception of depth [11]. This is possible with separate and slightly different displays for each eye that replicate sight in physical environments [11]. XR displays with high resolutions activate a wide user field of view that can span from 90 to 180 degrees. XR systems also offer superior auditory experiences in comparison to 2D systems. 3D, spatial or binaural audio allows the construction of soundscapes that enhance immersion decisively in AR and VR [12]. The spatial distribution of sound allows users to orientate themselves and identify the directions of sound cues, a powerful medium for navigation and user attention attraction.

In addition to the above passive sensory inputs, XR systems allow active interaction with virtual elements through the use of motion controllers. These are handheld input devices with a grip, buttons, triggers and thumbsticks. Using the controllers, users can touch, grab, manipulate and operate virtual objects [13]. This capability renders them active agents in any educational experience. On this front, the development of full hand tracking will further improve the user experience toward a more natural interface. Research is also being conducted towards wearable devices such as haptics suits and gloves that respond to touch [13]. Further sensory research efforts are concentrated in the direction of smell digitalization and simulation [14].

Interaction in XR environments does not require users to be stationary. Users can activate their entire bodies. Physical movement is being transferred into XR environments through positional and rotational tracking [15]. Movement can be tracked with either external, permanently mounted cameras (outside-in) or through inherent headset sensors and cameras that monitor position changes in relation to the physical environment (inside-out). The latter is used in stand-alone, wireless headsets. The supported degrees of freedom (DoF) of a XR headset is an essential specification that reflects its motion tracking capabilities [15]. Early and simpler headsets support three rotational head movement DoFs. Contemporary and high-fidelity headsets support all six DoFs adding lateral body movement along the x, y and z axes [15]. One frontier pertaining to occluded VR spaces is perpetual movement translation through unidirectional treadmills [16].

## 3. Limitations of 2D Learning Environments

Online distance education has a long history associated with the movement and philosophy of Open Education. The Open Education movement led to the creation of Open Universities worldwide, mainly after the 1960s [17]. Later, Computer Science advancements and the Internet enabled the emergence of Open Courseware, Open Educational Resources and Open Educational Practices [18]. More recently, it triggered the explosion of Massive Open Online Courses (MOOCs). MOOCs are openly accessible online courses that are attended by hundreds or thousands of people. Most of the time, they have a duration of a few weeks and are free of charge [19].

Online learning is becoming increasingly mainstream especially in higher and adult, continuous education. The COVID-19 pandemic accelerated this trend by disrupting attendance-based activities in all levels of education. Remote emergency teaching was enforced worldwide due to health-related physical distancing measures [20]. Ever since its conception, online education mainly relies on two main system types: Asynchronous and synchronous e-learning [21]. Both types depend on software or web applications in two-dimensional digital environments, spanning in-plane digital windows with width and height but without any depth. Standard asynchronous online learning tools include learning management systems (e.g., Moodle, Blackboard), and sometimes also collaborative web applications and social networks. Asynchronous tools serve the flexible, in other words, anytime, anywhere communication and interaction among educators, students and content [21]. Synchronous e-learning systems enable the online meeting of educators and students at the same time in a digital, virtual space. Synchronous online learning is implemented through web conferencing platforms (e.g., Zoom, WebEx, Microsoft Teams, Adobe Connect, Skype) [22].

However, applications operating in 2D, web-based environments have well-documented limitations and inefficiencies. The daily extended use of synchronous online platforms leads to phenomena such as Zoom fatigue [23]. Asynchronous platforms are often plagued by emotional isolation, a detrimental emotion for participation motivation. Consequently, e-learning courses in the above-mentioned platforms face high drop-out rates [19]. This phenomenon reaches its extreme in MOOCs where typical completion rates have been fluctuating around or below 10% [24]. The use of social media and collaborative applications (e.g., blogs, wikis) can improve active engagement but not necessarily address

natural communication and users' emotional stress [25]. 2D platforms have the following limitations that impact education negatively:

- Low self-perception: Users experience a very limited perception of the self in 2D environments. They are represented as disembodied entities through a photo or a live webcam head shot feed with no personalization options.
- No presence: Web conferencing sessions are perceived as video calls to join rather than virtual collective meeting places. Participants in long meetings tend to lean out and be distracted.
- Inactivity: 2D platforms offer limited ways of interaction among participants. Unless instructors initiate a learning activity, students are confined to passive participation with few opportunities to act.
- Crude emotional expression: Users have very limited options to express their feelings through smileys and emojis.

All these limitations can be addressed with 3D, immersive spatial environments.

## 4. Brief History of Virtual Media and XR Technologies

VR usually brings to mind futuristic science fiction images and sophisticated hardware. However, it is essential to realize that VR is associated with procedures in the human brain that do not require any equipment. Humans can experience an alternate reality through imagination as a thought, fantasy or mind-wandering. In fact, virtual-world building is an essential part of the human experience from the primordial, distant first days of the human species. Blascovich and Bailenson provide a detailed timeline of human virtual communication media that served this transportation of conscience [26]. Having a clear and broader comprehension of the past of virtual media is essential to articulate creative future visions and innovative solutions to complex problems with immersive technologies. This knowledge is fundamental for future Metaverse applications.

Prehistoric cave paintings and oral storytelling were the first media to capture and immortalize tribe tales. They were used to build virtual, mythological worlds to communicate both real and allegoric events, as well as valuable lessons learned to the plenary and the next generations. In ancient Greece, theater was a community-centered institution that transferred audiences to historical or mythological places and times. Theatric plays, both tragedies and comedies, constituted a social procedure shaping the collective identity, instigating political discourse and action. Plato's allegory of the cave illustrated the perception of the world based on our internal mental models and the dichotomy between the physical and virtual realities [27]. In medieval times, handwritten manuscripts and book reproduction through typography opened new horizons to human thought that fueled the Renaissance [26].

Later, analog inventions such as photography, cinematography, electricity, the telephone and mass media, e.g., radio and television, allowed the construction of virtual realities on a mass scale. One famous example is Orson Welles' highly realistic radio drama broadcast of excerpts from H.G. Wells' book "The war of the worlds" in 1938. Thousands of citizens believed the fictional story, panicked and fled in fear of an eminent alien invasion, an early mass media fake news hysteria [28].

In the modern era, the Link Trainer is the first analog precursor to VR, a mechanical flight simulator [29]. It was used in the late 1920s to train large cohorts of military airplane pilots. In the 1960s, the first multisensory systems were developed. Morton Heilig's Sensorama machine was a public, stand-alone arcade machine that provided immersive, multimodal theatrical experiences for entertainment [30]. More specifically, players could experience a simulated tour with a motorbike in city streets through an occluded screen movie, a vibrating seat, sounds, an odor transmitter and wind fans. In 1968, the first experimental, mechanical AR heads-up-display was developed by Ivan Sutherland. It earned the nickname Sword of Damocles, due to the fact that it was quite heavy, mounted and hanging from the ceiling [4]. In the 1980s, Myron Krueger introduced the term Artificial Reality. His innovative installation Videoplace demonstrated how remote real-time interactions

were possible in computer-generated environments [31]. Moreover, the first commercial VR applications appeared and the term VR was coined by Jaron Lanier [32]. Bulky immersion helmets became portable and tethered with peripheral haptic devices such as gloves and even body suits. During the 1990s, the cave automatic virtual environment (CAVE) room-scale VR system was developed. This was an example of semi-immersive VR through partial visual and operation realism. Stereoscopic images were projected onto the walls of a room to create depth in the users' visual field, which in turn provides a sense of partial immersion. This technology is widely adopted for art and museum installations. Furthermore, interconnected networks and the Internet enabled the emergence of the multiuser, social, computer-based, non-immersive VR platforms or virtual worlds. This trend was accelerated and massively adopted in the 2000s. In the 2010s, computer science advancements led to the development of the first affordable VR head-mounted displays or headsets, Oculus Rift, HTC Vive and Google Cardboard, propelling immersive VR to the next technological level towards becoming mainstream. In the 2020s, consumer-grade wireless, stand-alone VR headsets are the norm. Enterprise-grade MR headsets e.g., Microsoft HoloLens, Magic Leap and AR wearable smart glasses have also emerged [33].

## 5. Virtual Worlds and Virtual Reality in Education

VR technologies initially offered single-user experiences since networking computing was in its infancy. Computer networks allowed the ascension of collective, social non-immersive VR spaces named virtual worlds. A virtual world is a persistent, computer-generated networked environment where users meet and communicate with each other just they would in a shared space [34]. In the first years of networked computing, during the late 1970s, the first generation of social VR systems was text-based. They were called Multi-User Dungeons (MUDs), role-playing games in fantasy settings where players choose avatars from different classes to develop specific skills or powers, explore or complete quests [35]. MUDs were inspired by the role-playing board game Dungeons & Dragons and Tolkien's masterful fantasy works Hobbit and Lord of the Rings [36]. In 1989, Habitat was the first virtual world platform with a 2D graphical interface. The second wave of social VR systems followed in the 1990s and 2000s where platforms such as Traveler, Croquet, Active-Worlds, There, Blue Mars, Second Life and Open Simulator used client-server architecture and integrated a graphical user interface and multimedia communication [37]. Second Life has achieved remarkable longevity since 2003 thanks to economical sustainability around its virtual economy, currency and programming capabilities. It is still active today thanks to a vibrant community of returning creators and users. A new, third generation of social VR environments offering sensory immersion include VRChat, AltSpaceVR, EngageVR, RecRoom, Virbela, Sansar, High Fidelity, Sinespace, Somnium Space, Mozilla Hubs, Decentraland, Spatial and Meta (formerly known as Facebook) Horizon Worlds [38]. These platforms offer an embodied user representation and a series of tools for online education and remote meetings. Some of them, e.g., RecRoom, Virbela, allow access and participation through multiple devices beyond VR HMDs, such as desktop systems and mobile applications.

### 5.1. VR Affordances

Rosenblum and Cross [39] mention three essential characteristics in all VR systems: Immersion, interaction and optical fidelity. Immersion determines the degree that the user feels that s/he is cognitively teleported to an alternative, synthetic world. There are two qualities of immersion: Socio-psychological immersion and multimodal immersion. Socio-psychological immersion is a universal experience as it can be achieved with multiple ways and means. Users of any medium such as a book, radio or television feel that they are transported into a remote or imaginary place that is created mentally due to the received mediated information. Multimodal immersion requires sophisticated equipment such as VR headsets and haptic suits that provide information to the brain through sensory channels such as sight, sound, touch and smell. Interaction is also twofold: With the

manipulation, creation and modification of virtual objects and with other people when they co-exist in the same virtual space [15]. Visual or representative fidelity can lead to the suspension of disbelief, a feeling that users are in an entirely synthetic, artificial space.

Dalgarno and Lee [40] argue that immersion with interaction in 3D virtual worlds leads to the additional affordances of identity construction, presence and co-presence. The construction of an online identity is achieved primarily through the digitally embodied representation of the self in virtual worlds, the avatar. Avatar is a word in the Sanskrit language signaling the manifestation of a deity in human form. Analogously, in social VR environments, each user materializes and is visible as a digital agent, persona or avatar. This is a key difference in comparison to 2D web-conferencing platforms. Avatars enable a superior sense of self since participants control their avatars [41]. Avatars' characteristics can be personalized meticulously to reflect users' self-expression freedom; they can appear in human-like or completely fantastical form [37]. The identification with one's avatar in a virtual environment can have profound psychological impact on behavior and learning; embodied experiences as avatars in virtual reality spaces have a direct influence on human behavior and transfer to the physical world. This phenomenon is called the Proteus effect [42].

The embodied digital identity and the ability to engage with the environment and virtual objects in multiple points of view, such as the third-person perspective, creates the psychological sense of being in a space, experiencing presence [37]. Presence or telepresence is the perceptual illusion of non-mediation [43]. The psychological illusion of non-mediation implies that users fail to perceive the existence of a medium that intervenes in their communicative environment [44]. Consequently, they act though this technological medium is nonexistent. Presence is extended through communication with other people [45]. Co-presence is the feeling of being together in a virtual space. Meeting synchronously in the same 3D virtual space with other avatars and acknowledging the persons behind them leads to experiencing a prevalent power of co-presence. Co-presence is essential for education and the construction of virtual communities of practice and inquiry [46].

*5.2. VR in Education*

Bailenson [47] suggests the integration of immersive, headset-based VR into education for four primary purposes or use case scenarios in location-based education. First, for rehearsing and practicing dangerous activities such as piloting an airplane or conducting a surgical operation where the stakes of failure are very high with grave consequences. Second, to reenact an inconvenient or counterproductive situation such as managing a problematic behavior in a school or handling a demanding business client. A third use case is to perform something impossible such as internal human body organs observation or to travel virtually back in time to archeological site reconstructions. Fourth, immersive VR is also recommended for rare or too expensive experiences such as a group field trip to a tropical forest or an underwater wreckage. All the above use cases can be classified as single-user experiences.

Bambury [48] takes into account the social aspect in VR and discerns four stages and respective aims of VR pedagogical implementation into teaching. The first stage is using VR for perception or stimulation. The educator directs a multimodal experience and students follow in a passive role, e.g., seeing a 360 or spherical video in Youtube VR. In the second stage, students can have basic interactions with the virtual world and influence the content's flow. The achievement of autonomy is the next state. Students have a high level of autonomy and guide the learning procedure with their decisions as demonstrated in Google Earth VR. Finally, the ultimate stage is presence. Students have a genuine sense of being in a new space through human interaction and collaboration in social VR platforms.

Collective VR spaces enable the wider application of blended active learner-centered instructional design strategies such as problem-, project and game-based learning [49,50]. Online learning in social VR allows the wider deployment of game-based learning methods.

These motivation amplification methods include playful design, gamification and serious games that can be applied in the micro-, meso- or macro-level of an online course [51]. The macro-level encompasses the entire course design and its evaluation. The meso-level comprises a unit or a specific procedure of a course. For instance, course communication messages could be expressed playfully with linguistic terms, metaphors and aesthetic elements from a specific domain and theme that can be of relevance to the subject matter e.g., sailing or mountain climbing. One class meeting or learning activity (e.g., an assignment) constitutes the micro-level. Playful design can be utilized to foster a relaxed and creative learning culture of inclusion, initiative and experimentation [52]. Entire online courses can be gamified and organized as multiuser online games in virtual worlds [53]. Complex epistemic games based on realistic simulations and escape rooms can be developed in VR as complementary resources and practice activities [54]. Simulations and gameful experiences in VR provide opportunities for learners to apply theoretical knowledge, experiment with equipment, practice complex procedural and behavioral skills and learn from their mistakes without the gravity of the consequences or errors in the physical world [5].

## 6. Metaverse Contemporary Development

### 6.1. Literary Origin

The term Metaverse was invented and first appeared in Neal Stevenson's science fiction novel Snow Crash published in 1992 [55]. It represented a parallel virtual reality universe created from computer graphics, which users from around the world can access and connect through goggles and earphones. The backbone of the Metaverse is a protocol called the Street, which links different virtual neighborhoods and locations an analog concept to the information superhighway. Users materialize in the Metaverse in configurable digital bodies called avatars. Although Stevenson's Metaverse is digital and synthetic, experiences in it can have a real impact on the physical self. A literary precursor to the Metaverse is William Gibson's VR cyberspace called Matrix in the 1984 science fiction novel Neuromancer [36].

A modern literary reincarnation of the Metaverse is the OASIS, illustrated in the 2011 science fiction novel Ready Player One authored by Ernest Cline [56]. OASIS is a massively multiuser online VR game that evolved into the predominant online destination for work, education and entertainment. It is an open game world, a constellation of virtual planets. Users connect to OASIS with headsets, haptic gloves and suits. Regarding education, OASIS is much more than a public library containing all the worlds' books freely and openly accessible to citizens. It presents a techno-utopic vision of virtual online education. Hundreds of luxurious public-school campuses are arranged in the surface of a planet dedicated exclusively to K-12 education. Online school classes are superior in comparison to the grass-and-mortar schools as they resemble holodecks: Teachers take students for virtual field trips to ancient civilizations, foreign countries, elite museums, other planets and inside the human body. As a result, students pay attention, are engaged and interested.

### 6.2. Metaverse Implementations

In the field of VR, the Metaverse was conceived as the 3D Internet or Web 3.0 [57]. Its first iteration was conceived as a web of virtual worlds where avatars would be able to travel seamlessly among them. This vision was realized in Opensim's Hypergrid [36]. Different social and stand-alone virtual worlds based on the open-source software Opensimulator were—and still are—reachable through the Hypergrid network that allows the movement of digital agents and their inventory across different platforms through hyperlinks. However, Hypergrid was and is still not compatible with other popular proprietary virtual worlds such as Second Life.

Currently, the second MR iteration of the Metaverse is under construction where social, immersive VR platforms will be compatible with massive multiplayer online video games, open game worlds and AR collaborative spaces. According to this vision, users can meet, socialize and interact without restrictions in an embodied form as 3D holograms

or avatars in physical or virtual spaces. Currently, this is possible with several limitations within the same platform. Cross-platform and cross-technology meetings and interactions, where some users are in VR and others in AR environments, are the next frontier. Common principles of the Metaverse include software interconnection and user teleportation between worlds. This necessitates the interoperability of avatar personalization and the portability of accessories, props and inventory based on common standards. The seven rules of the Metaverse comprise a high-level manifesto, a proposal for future development based on previously accumulated experience with the development of the Internet and the World-Wide Web [58]. According to this proposal, there should be only one Metaverse (rule #1), and not many Metaverses or Multiverses, as the next iteration of the Internet (rule #7). As such, the Metaverse should be for everyone, (#2) open (#4), hardware-agnostic (#5), networked (#6) and collectively controlled (#3) [58].

### 6.3. Metaverse Challenges

The Metaverse faces a number of challenges related to the underlying AR and VR technologies. Both technologies are persuasive and can influence users' cognition, emotions and behaviors [59]. The high cost of equipment is a barrier to mass adoption that is expected to be mitigated in the long run. Risks related to AR can be classified into four categories related to (i) physical well-being, health and safety, (ii) psychology, (iii) morality and ethics and (iv) data privacy [60]. On the physical level, the attention distraction of users in location-based AR applications has led to harmful accidents. Information overload is a psychological challenge that needs to be prevented. Moral issues include unauthorized augmentation and fact manipulation towards biased views. Data collection and sharing with other parties constitutes the risk with the widest implications in regards to privacy [60]. The additional data layer can emerge as a possible cybersecurity threat. Volumetric capturing and spatial doxxing can lead to privacy violations. More importantly, Metaverse actors can be tempted to compile users' biometric psychography based on user data emotions [61]. These profiles could be used for unintended behavioral inferences that fuel algorithmic bias.

Related to VR, motion sickness, nausea and dizziness are among the most commonly reported health concerns [5]. Head and neck fatigue is also a limitation for longer use sessions due to VR headsets' weight. Extended VR use could lead to addiction, social isolation and abstinence from real, physical life, often combined with body neglect [59]. Another known drawback of open social worlds is toxic, antisocial behavior, e.g., griefing, cyber-bullying and harassment [62]. High-fidelity VR environments and violent representations can trigger traumatic experiences. Related to data ethics, artificial intelligence algorithms and deep learning techniques can be utilized to create VR deep fake avatars and identity theft.

### 6.4. Meta-Education

Pertaining to Metaverse's potential for educational radical innovation, laboratory simulations (e.g., safety training), procedural skills development (e.g., surgery) and STEM education are among the first application areas with spectacular results in terms of training speed, performance and retention with AR and VR-supported instruction [9,63,64]. Thanks to the ability to capture 360-degree panoramic photos and volumetric spherical video, the Metaverse can enable immersive journalism to accurately and objectively educate mass audiences on unfamiliar circumstances and events in remote locations [65]. Moreover, new models of Metaverse-powered distance education can emerge to break the limitations of 2D platforms. Meta-education can allow rich, hybrid formal and informal active learning experiences in perpetual, alternative, online 3D virtual campuses where students are co-owners of the virtual spaces and co-creators of liquid, personalized curricula.

### 6.5. Innovative Immersive Metaverse Applications

In the context of the Metaverse, immersive technologies will have further applications in the fields of Spatial Computing and Brain–computer interface [66]. Spatial Computing

allows the control of computing equipment with natural gestures and speech [67]. Brain–computer interfaces enable communication with computing devices exclusively through brain activity [68] for the control of a synthetic limb or to empower paralyzed persons to operate computers. Moreover, the integration of blockchain-based crypto-currencies (e.g., bitcoin) and non-fungible tokens (NFTs) allow the deployment of innovative virtual economy transactions and architectures [69]. On a broader scale, Metaverse-related technologies are expected to cross-pollinate, expand and be further amplified by exponential technologies such as wireless broadband networks, cloud computing, robotics, artificial intelligence and 3D printing. All these technologies mark a transition to a fourth industrial revolution. In other words, the Metaverse is expected to be an important aspect of Education 4.0, education in the Industry 4.0 era [70].

## 7. Conclusions

The Metaverse is not a new concept. Its main dimensions are illustrated in Figure 1. However, in the context of MR, it can bridge the connectivity of social media with the unique affordances of VR and AR immersive technologies. If the interplay among them is unleashed creatively, it promises to transform many industry sectors, among them distance online education. New models of Meta-education, Metaverse-powered online distance education, can emerge to allow rich, hybrid formal and informal learning experiences in online 3D virtual campuses. Online learning in the Metaverse will be able to break the final frontier of social connection and informal learning. Physical presence in a classroom will cease to be a privileged educational experience. Telepresence, avatar body language and facial expression fidelity will enable virtual participation to be equally effective. Additionally, social mixed reality in the Metaverse can enable blended active pedagogies that foster deeper and lasting knowledge [71]. More importantly, it can become a democratizing factor in education, enabling world-wide participation on equal footing, unbound by geographical restrictions.

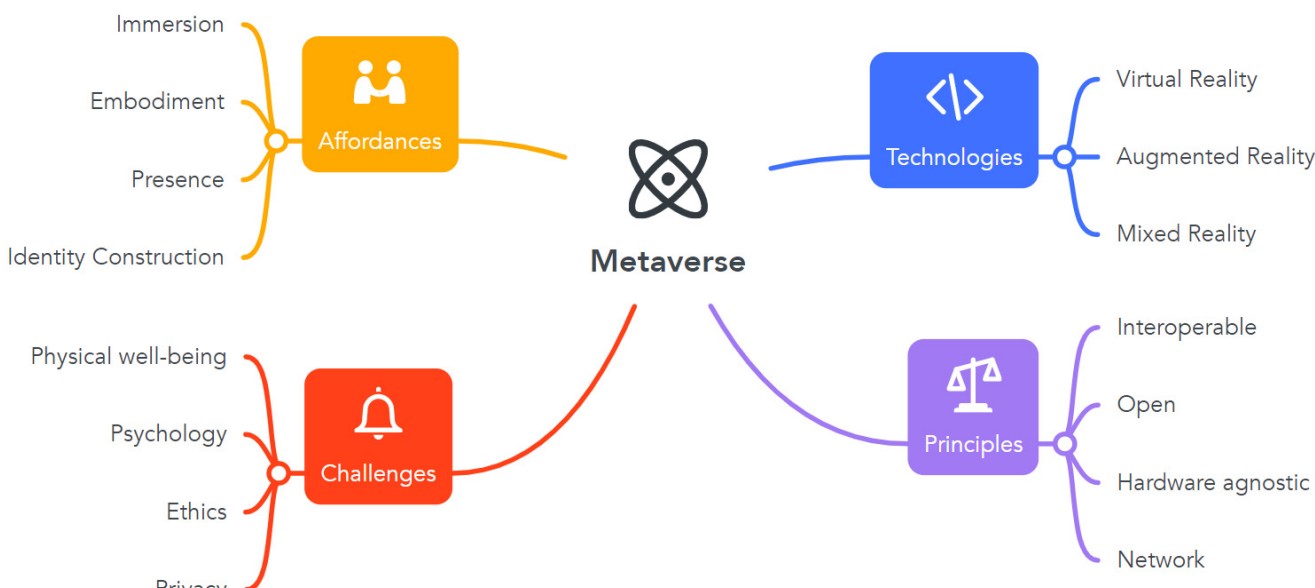

**Figure 1.** Metaverse technologies, principles, affordances and challenges.

**Funding:** This research received no external funding.

**Conflicts of Interest:** The author declares no conflict of interest.

**Entry Link on the Encyclopedia Platform**

https://encyclopedia.pub/20619.

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
