# Peer review of "Metaverse"

_encyclopedia, doi:10.3390/encyclopedia2010031_

Round 1

Reviewer 1 Report

Well done. I hav no suggested changes.

Author Response

Thank you for your kind comment.

Reviewer 2 Report

The manuscript offers an overview of types of artificial reality, leading to a definition and positioning of the term metaverse. It is timely and relevant. However I have a few concerns with the manuscript, and a few suggestions for improvement.

First, the manuscript knows many language and typing errors. Verbs and letters are missing, and words are inconsistently used (for instance step, stage, and level indicating parts of the same development). Orson Wells should be Orson Welles. Please check and correct.

Next, the current organisation of the manuscript puts a high burden on the reader's cognitive load: it is not until section 6 (of 7) that the central notion of Metaverse is introduced and explained. I would suggest to consider a different structure by first either defining - or briefly sketching - Metaverse, and then unravel its origins. Also, editing the text for paragraphs to include one - and only one - idea or concept is suggested. And the text could be improved with better bridges between paragraphs and even sentences. This would improve readability, and in my view audience reach.

Think of readers two or three years from now. Would the opening still be relevant then? I do not think so. The development of online learning tools has accelerated over the last few years, but it only sped up a process that was already happening. Also, the idea of limitations and inefficiencies of current '2D' tools as a premisse for the article could be presented more in-depth and made stronger.

Section 3 in my view misses reference to 'Artificial Reality' as coined by Myron Krueger, and operationalised in 'Videoplace'. For one thing, Krueger's work highly influenced Jaron Lanier. Second, it would fit the references to literature lateron in the text.

4.1: I don't understand why the heading includes the word 'Social'. This calls for discussing other types of affordances as well (see for an overview Kirschner 2004). The same counts for the heading of section 5.

Author Response

The manuscript offers an overview of types of artificial reality, leading to a definition and positioning of the term metaverse. It is timely and relevant. However I have a few concerns with the manuscript, and a few suggestions for improvement.

R2C1: First, the manuscript knows many language and typing errors. Verbs and letters are missing, and words are inconsistently used (for instance step, stage, and level indicating parts of the same development). Orson Wells should be Orson Welles. Please check and correct.

The manuscript has undergone proof-reading and all identified grammatical and syntax errors have been corrected. The concepts step, stage and level were used deliberately in section 5 as alternate concepts - synonyms to avoid repetition. As suggested, I revised the expressions for improved consistency and to avoid any confusion.

R2C2: Next, the current organisation of the manuscript puts a high burden on the reader's cognitive load: it is not until section 6 (of 7) that the central notion of Metaverse is introduced and explained. I would suggest to consider a different structure by first either defining - or briefly sketching - Metaverse, and then unravel its origins.

The organization of the manuscript has been revised. A new introduction has been authored and the central concept of the Metaverse is mentioned and explained on page 1.

R2C3: Also, editing the text for paragraphs to include one - and only one - idea or concept is suggested. And the text could be improved with better bridges between paragraphs and even sentences. This would improve readability, and in my view audience reach.

The manuscript’s language has been carefully reviewed and revised according to the suggestions to improve its readability.

R2C4: Think of readers two or three years from now. Would the opening still be relevant then? I do not think so. The development of online learning tools has accelerated over the last few years, but it only sped up a process that was already happening.

A new, revised Introduction has been authored that will hopefully be more relevant for readers in the future.

R2C5: Also, the idea of limitations and inefficiencies of current '2D' tools as a premisse for the article could be presented more in-depth and made stronger.

Thank you for the suggestion. A new section 2 on the limitations of 2D Learning Environments has been added.

R2C6: Section 3 in my view misses reference to 'Artificial Reality' as coined by Myron Krueger, and operationalised in 'Videoplace'. For one thing, Krueger's work highly influenced Jaron Lanier. Second, it would fit the references to literature lateron in the text.

Thank you for this insightful remark. The following text was added: Myron Krueger’s innovative installation Videoplace demonstrated how remote real-time interactions were possible in computer-generated environments and introduced the term Artificial Reality [17].

R2C7: 4.1: I don't understand why the heading includes the word 'Social'. This calls for discussing other types of affordances as well (see for an overview Kirschner 2004). The same counts for the heading of section 5.

Thank you for this observation. The titles of both sections have been revised.

Reviewer 3 Report

Dear Author, 

The paper presents the definition of the Metaverse Social Mixed Reality. The title creates expectation as this is a subject gaining traction by the recent growth of XR technologies. In the introduction you set the context of the article towards education and how the traditional teaching tools for remote learning lack in comparison to XR. However, this is underexplored and it requires more information to properly understand the limitations and how XR can make it better. There are aspects of section 2 that are fundamental to the introduction to better understand the case being presented. With respect to section 3, it is unclear the association between the media being discussed and VR. This is of particular importance as the media being discussed is similar to the media and tools criticized in the introduction. Additionally, VR is used inconsistently with respect to the provided definition. I suggest changing these three paragraphs to discuss how the human visual, auditory, and haptic system work as these provide better grounds to understand how XR continues disrupting how teaching is conducted. This also allows to make stronger connection between the immersive technologies and fundamental concepts such as immersion, presence, and embodiment. Furthermore, this will allow to explain why web-based or text-based computer application lack in terms of hands-on activities with XR. Given the current spike on XR and social applications, I think discussing current works, trends and challenges is more important that connections to pop culture and literary works. However, given the social components, a section could discuss managing expectation from these media to reality.

Finally, the conclusions fall short on delivering the paper's outcome as the origin of the metaverse was not entirely clear, here I suggest creating a diagram that integrates the metaverse alongside XR and the virtuality continuum. Regarding the affordances, some were introduced, but I think the challenges and drawbacks need to be discussed as well, and with respect to the transformation of education, business, work, and entertainment, more information is needed to understand why is this the case.

Overall, the paper is well presented, minor grammar corrections and review is needed.

Author Response

Dear Author, 

The paper presents the definition of the Metaverse Social Mixed Reality. The title creates expectation as this is a subject gaining traction by the recent growth of XR technologies.

R3C1: In the introduction you set the context of the article towards education and how the traditional teaching tools for remote learning lack in comparison to XR. However, this is underexplored and it requires more information to properly understand the limitations and how XR can make it better.

Thank you for this suggestion. A new section 2 on the limitations of 2D Learning Environments has been added.

R3C2: There are aspects of section 2 that are fundamental to the introduction to better understand the case being presented.

A new, revised Introduction has been authored where the central concept of the Metaverse is mentioned.

R3C3: With respect to section 3, it is unclear the association between the media being discussed and VR. This is of particular importance as the media being discussed is similar to the media and tools criticized in the introduction. Additionally, VR is used inconsistently with respect to the provided definition.

The first paragraph of the section has been revised and expanded to make the connection clearer between VR and virtual media, ensuring the consistent use of term VR. Moreover, the following text has been added to signal its importance: “Having a clear and broader comprehension of the past of virtual media is essential to articulate creative future visions and innovative solutions to complex problems with immersive technologies. This knowledge is fundamental for future Metaverse applications.”

R3C4: I suggest changing these three paragraphs to discuss how the human visual, auditory, and haptic system work as these provide better grounds to understand how XR continues disrupting how teaching is conducted. This also allows to make stronger connection between the immersive technologies and fundamental concepts such as immersion, presence, and embodiment. Furthermore, this will allow to explain why web-based or text-based computer application lack in terms of hands-on activities with XR.

A new, comprehensive subsection (8.2) on multimodal interactions in the Metaverse has been added.

R3C5: Given the current spike on XR and social applications, I think discussing current works, trends and challenges is more important that connections to pop culture and literary works. However, given the social components, a section could discuss managing expectation from these media to reality.

A separate section is dedicated to Metaverse’s literary origin story. Section 8 on contemporary Metaverse development issues and challenges has been reorganized and expanded.

R3C6: Finally, the conclusions fall short on delivering the paper's outcome as the origin of the metaverse was not entirely clear, here I suggest creating a diagram that integrates the metaverse alongside XR and the virtuality continuum.

A new, synoptic diagram on the Metaverse has been added to the paper.

R3C7: Regarding the affordances, some were introduced, but I think the challenges and drawbacks need to be discussed as well, and with respect to the transformation of education, business, work, and entertainment, more information is needed to understand why is this the case.

A new subsection (8.3) on the challenges of the Metaverse has been authored and added.

R3C8: Overall, the paper is well presented, minor grammar corrections and review is needed.

Thank you for the kind comment, the manuscript has undergone proof-reading and all identified grammatical and syntax errors have been corrected.

Round 2

Reviewer 2 Report

Thank you for considering the reviewers’ comments. In my view the readability, scientific relevance, and practical relevance have improved. Also, the new introduction and premisse has increased the manuscript’s lifespan.

Author Response

Thank you for your kind feedback.

Reviewer 3 Report

Dear Authors, 

The manuscript introduced improvements based on previous comments. While these clarifications and additions enhanced the quality of the manuscript, English grammar remains a main concern. Additionally, the manuscript indicates that the metaverse centers around education. However, the way the concepts are presented does not reflect this when presenting the Metaverse. Furthermore, a section is devoted to metaverse in science fiction, rather than presenting the metaverse in education. Applications and developments are general with little connecting tissue to education. 

Consider a different workflow that helps build the contents from a general overview to the specificity of the discussion being presented. For example, start with section 3, then a mix of 2 and 4, then 5 and 6 as one, and the remainder as is. Please note that section 7 is missing.

The introduction discusses a race occurring to develop the Metaverse. However, the running argument here was the race to construct the metaverse, but none of these sections seem to touch base on that. This is disconnected and can provide valuable information to understand how it impacts education.

In section 2, it is unclear what is the connection between online education and open education? Why is it matter those proponents (who?) suggest that education is a human right and how does that connect to the topic of the metaverse? Furthermore, the argument for open education lacks more discussion with respect to the Metaverse as there are several entry-level barriers including affordability, inclusive design, and others.

When reviewing the history of VR, the Link Trainer is referred to be the first VR application, but this contradicts the definition provided and the partition of the term for the first time.  Additionally, fundamental components of immersive experiences such as depth perception, stereo vision, and haptics are not introduced. This is confusing as these terms are later employed, thus assuming the reader's familiarity with them. 

When discussing the advances in immersive technologies, in particular VR headsets, the authors stopped in 2010. Technology has changed since then and there aren't any mentions of the state of the technology in 2022. When discussing VR platforms, the authors list several without any analysis to understand how these have evolved and how they contribute to the Metaverse.

When discussing the Metaverse, seven rules are cited, but not explained. Additionally, MR is treated as the true technology for the Metaverse and presented as something that is already happening. However, the discussion sees this as an expectation, which contradicts earlier statements. 

The figure quality has improved, but it is lacking details. For example, MR plays a significant role, but it is not included within the technologies.  The caption of Figure 1 has a bad break, resulting in being placed on a different page.

Finally, additional comments and suggestions are included in the PDF.

Author Response

Dear reviewer,

Thank you for giving me the opportunity to revise the entry entitled “Metaverse Social Mixed Reality” (Manuscript ID: 1552947). I would like to inform you that the changes requested have been addressed to the best possible degree and the manuscript has been expanded significantly. I hereby provide a point-by-point report detailing how the changes have been made. With brown colour I have included the comments received from the reviewers. With blue font are the corrections, changes, and additions made in the revised manuscript. All changes made can be observed (with the track changes feature) in the attached, revised manuscript version.

Reviewer 3

R3C1. The manuscript introduced improvements based on previous comments. While these clarifications and additions enhanced the quality of the manuscript, English grammar remains a main concern. Additionally, the manuscript indicates that the metaverse centers around education. However, the way the concepts are presented does not reflect this when presenting the Metaverse. Furthermore, a section is devoted to metaverse in science fiction, rather than presenting the metaverse in education. Applications and developments are general with little connecting tissue to education. Consider a different workflow that helps build the contents from a general overview to the specificity of the discussion being presented. For example, start with section 3, then a mix of 2 and 4, then 5 and 6 as one, and the remainder as is. Please note that section 7 is missing.

A new workflow of the manuscript was constructed to address this issue. Prior section 3 was moved up. Sections 5 and 6 were merged as well as sections 7 and 8.

R3C2. The introduction discusses a race occurring to develop the Metaverse. However, the running argument here was the race to construct the metaverse, but none of these sections seem to touch base on that. This is disconnected and can provide valuable information to understand how it impacts education.

The introduction has been expanded to set with improved accuracy the main argument around a new model of online education.

R3C3. In section 2, it is unclear what is the connection between online education and open education? Why is it matter those proponents (who?) suggest that education is a human right and how does that connect to the topic of the metaverse? Furthermore, the argument for open education lacks more discussion with respect to the Metaverse as there are several entry-level barriers including affordability, inclusive design, and others.

Open Education was introduced to establish briefly the history and state in the field of distance online education. As requested, information not directly linked with the Metaverse were omitted to avoid any confusion.

R3C4. When reviewing the history of VR, the Link Trainer is referred to be the first VR application, but this contradicts the definition provided and the partition of the term for the first time. 

Corrected. The Link Trainer is cited as “the first analog precursor to VR”.

R3C5. Additionally, fundamental components of immersive experiences such as depth perception, stereo vision, and haptics are not introduced. This is confusing as these terms are later employed, thus assuming the reader's familiarity with them.

These components were introduced in Section 8.2 but were now moved up to Section 2.1, prior to any other subsequent references in the manuscript.

R3C6. When discussing the advances in immersive technologies, in particular VR headsets, the authors stopped in 2010. Technology has changed since then and there aren't any mentions of the state of the technology in 2022.

The following text was added: “In the 2020s, consumer-grade wireless, stand-alone VR headsets are the norm. Enterprise-grade MR headsets e.g. Microsoft HoloLens, Magic Leap and AR wearable smart glasses have also emerged”.

R3C7. When discussing VR platforms, the authors list several without any analysis to understand how these have evolved and how they contribute to the Metaverse.

The following text was added: “These platforms offer an embodied user representation and serious of tools for online education and remote meetings. Some of them, e.g. RecRoom, Virbela, allow access and participation through multiple devices beyond VR HMDs, such as desktop systems and mobile applications.”

R3C8. When discussing the Metaverse, seven rules are cited, but not explained.

Additional text was added: “The seven rules of the Metaverse comprise a high-level manifesto, a proposal for future development based on previously accumulated experience with the development of the Internet and the World-Wide Web. According to this proposal, there should be only one Metaverse (rule #1), and not many Metaverses or Multiverses, as the next iteration of the Internet (rule #7). As such the Metaverse should be for everyone, (#2) open (#4), hard-ware-agnostic (#5), networked (#6) and collectively controlled (#3).”

R3C9. Additionally, MR is treated as the true technology for the Metaverse and presented as something that is already happening. However, the discussion sees this as an expectation, which contradicts earlier statements. 

The later sentence in the discussion was omitted to avoid any contradiction or confusion.

R3C10. The figure quality has improved, but it is lacking details. For example, MR plays a significant role, but it is not included within the technologies.  The caption of Figure 1 has a bad break, resulting in being placed on a different page.

Figure 1 has been expanded to include MR. Its caption is on the same page.

R3C11. Finally, additional comments and suggestions are included in the PDF.

All suggestions have been carefully reviewed and corrected in the revised manuscript.

I would like to express my gratitude to the Editor and the Reviewers for their insightful suggestions and constructive comments that improved the validity and readability of this manuscript. I remain at your disposal for any further modifications that may be required.

The author

Round 3

Reviewer 3 Report

Dear Authors, 

Thank you for addressing the reviews. The current manuscript quality has increased significantly. The document is clear, concise, and provides an overview of the Metaverse and considerations for educational applications.

Author Response

Thank you indeed for your valuable feedback and positive comments.